# Methyl Jasmonate Treatment of Broccoli Enhanced Glucosinolate Concentration, Which Was Retained after Boiling, Steaming, or Microwaving

**DOI:** 10.3390/foods9060758

**Published:** 2020-06-08

**Authors:** Yu-Chun Chiu, Kristen Matak, Kang-Mo Ku

**Affiliations:** 1Division of Plant and Soil Sciences, West Virginia University, Morgantown, WV 26506, USA; yuchiu@mix.wvu.edu; 2Division of Animal and Nutritional Sciences, West Virginia University, Morgantown, WV 26506, USA; Kristen.Matak@mail.wvu.edu; 3Department of Horticulture, College of Agriculture and Life Sciences, Chonnam National University, Gwangju 61886, Korea

**Keywords:** broccoli, methyl jasmonate, glucosinolate, glucosinolate hydrolysis products, cooking

## Abstract

Exogenous methyl jasmonate (MeJA) treatment was known to increase the levels of neoglucobrassicin and their bioactive hydrolysis products in broccoli (*Brassica oleracea* var. *italica*), but the fate of MeJA-induced glucosinolates (GSLs) after various cooking methods was unknown. This study measured the changes in GSLs and their hydrolysis compounds in broccoli treated with MeJA and the interaction between MeJA and cooking treatments. All cooked MeJA-treated broccoli contained significantly more GSLs than untreated broccoli (*p* < 0.05). After 5 min of cooking (boil, steam, microwave), MeJA-treated broccoli still contained 1.6- to 2.3-fold higher GSL content than untreated broccoli. Neoglucobrassicin hydrolysis products were also significantly greater in steamed and microwaved MeJA-treated broccoli. The results show that exogenous MeJA treatment increases neoglucobrassicin and its hydrolysis compounds in broccoli even after cooking. Once the positive and negative effects of these compounds are better understood, the results of this experiment can be a valuable tool to help food scientists, nutrition scientists, and dieticians determine how to incorporate raw or cooked broccoli and *Brassica* vegetables in the diet.

## 1. Introduction

Broccoli consumption (*Brassica oleracea* var. *italica*) is associated with anti-cancer activity in in-vitro and in-vivo trials, due to the high content of phytochemicals, minerals, vitamins, and fibers. GSLs are a group of phytochemicals in broccoli which have been studied extensively for their health benefits; for example, an inverse relationship has been reported between the risk of breast cancer in Caucasian women and consumption of broccoli [1]. Furthermore, consumption of broccoli was shown to reduce the concentration of hepatic triglycerides in mice, and long-term broccoli consumption may promote liver health [2].

In *Brassica oleracea*, aliphatic GSLs derived from methionine and isoleucine are known [3,4], and indole GSL can be derived from tryptophan [3]. In plants, GSL are hydrolyzed by an endogenous enzyme myrosinase (EC 3.2.3.1, thioglucoside glucohydrolase) when insects attack or a physical wound is formed. This reaction produces the corresponding hydrolysis products. For example, isothiocyanates (ITCs) are GSL hydrolysis products reported to possess generally higher bioactivity than other types of GSL hydrolysis products [5]. The chemopreventive effect of sulforaphane, a well-known ITC, is greater than other GSL hydrolysis products in cell culture assay [6], animal assay [7], and a clinical trial [1]. However, the accumulation of its precursor, glucoraphanin, is mainly determined by the genetic background of the plant rather than the environmental factors [8]; therefore, increasing its concentration through agricultural practices is limited. In contrast, indole GSL levels are mainly affected by environmental factors [8], which implies that the concentration of indole GSL can be increased by agricultural practices.

Broccoli is often eaten cooked, or sometimes eaten raw as an ingredient in salads. Several studies have investigated the impact of common cooking methods, including boiling, steaming, and microwaving, on the retention of GSL, with boiling generally causing substantial loss of GSL [9,10,11]. In addition to the GSL loss, cooking also inactivates myrosinase, the enzyme converting GSL into hydrolysis products, and then hinders the formation of hydrolysis products [12]. To date, methods of delivering cooked broccoli without losing its nutritional benefits are still lacking in the literature, although consuming cooked broccoli is the most common practice for consumers.

Exogenous methyl jasmonate (MeJA) application increases inducible indole GSLs (neoglucobrassicin and glucobrassicin) or benzenic GSLs (gluconasturtiin) concentrations in *Brassica* vegetables [13] due to its anti-repellent activity. MeJA is a volatile jasmonate derivative that responds to herbivore damage [14,15], and GSL biosynthesis is upregulated by JA signaling transduction [16,17]. Applying exogenous MeJA triggers a defense response to insect damage in *Brassica* plants [18]; as a result, defense-related metabolites, such as GSLs, are upregulated [6] either by depletion of the carbon pool (glucose) or by the conserved transcriptional network [19]. MeJA application can be used as an agricultural practice to enhance the nutritional quality of broccoli. Indole-3-carbinol (I3C; derived from glucobrassicin) was associated with reducing the risk of hormone-responsive cancers [20] including breast [21] and prostate cancer [22]. *N*-methoxy-indole-3-carbinol (NMI3C), derived from neoglucobrassicin, was reported to reduce the growth of human colon cells by inhibiting CYP1A enzyme, and its potency was higher than I3C in vitro [23]. On the other hand, indole GSL was reported to have negative effects on health in previous literature [24,25]. It was shown that neoglucobrassicin is highly mutagenic [24], and the consumption of I3C (hydrolysis product of glucobrassicin) was suggested to have Janus properties, where I3C may promote carcinogenesis if it is ingested after the carcinogen in animal models [25]. Quantification of these compounds in raw and cooked *Brassica* crops may provide valuable information for nutrition scientists and dieticians once the effects of these compounds in humans becomes clearer.

Although MeJA treatment significantly increases GSL in broccoli, there has been limited study on myrosinase activity and the amount of GSL hydrolysis products in MeJA-treated broccoli after cooking. We hypothesized that MeJA-induced changes in broccoli may affect GSL degradation, myrosinase activity, or myrosinase-associated proteins during domestic cooking, resulting in different GSL or hydrolysis product compositions. As GSL hydrolysis compounds are the actual bioactive compounds rather than GSL itself [26], it is critical to determine whether MeJA treatment increases GSL hydrolysis products and myrosinase in cooked broccoli. Therefore, the major research questions of this study were to evaluate how MeJA application to broccoli plants will affect GSL concentration, myrosinase activity, GSL hydrolysis product amounts, and other metabolites in broccoli florets after boiling, steaming, and microwaving treatments.

## 2. Materials and Methods 

### 2.1. Broccoli Cultivation and Methyl Jasmonate Treatment

‘Green Magic’ broccoli was selected for its consistent response to 250 µM MeJA application year to year [27]. Seeds of ‘Green Magic’ broccoli were obtained from Johnny’s Selected Seeds (Albion, ME, USA). Broccoli was grown from seeds in the cell pack (one plant per pack) filled with Sunshine #1 Mix (Sun Gro Horticulture, Vancouver, BC, Canada) in the greenhouse (Morgantown, WV, USA). The seedlings were transplanted to 15-cm pots when the seedlings could be easily removed from the cell pack. After three weeks, seedlings with similar maturity and no sign of disease were then transplanted to 8.7-L pots to grow to full maturity. The greenhouse was normally under natural radiance with the capacity for high-pressure sodium (600 W HS200 deep reflector; Hortilux, Pijnacker, The Netherlands) supplemental lighting if the light intensity was below 50 W/m^2^. The temperature and supplemental lighting regime in the greenhouse were set at 25/18 °C and 14/10 h day/night. 

Forty untreated broccoli heads were harvested eight weeks after transplanting. Meanwhile, another 40 plants were treated with 250 µM MeJA in 0.1% aqueous Triton X-100 (Sigma-Aldrich, St. Louis, MO, USA) solution three days before harvesting. The harvested broccoli heads were stored at 4 °C prior to cooking.

### 2.2. Broccoli Preparation and Cooking Process

#### 2.2.1. Broccoli Sample Preparation

The harvested broccoli heads were chopped into 3–4-cm wide florets. After chopping, broccoli branchlets were randomized and were weighed 100 g for each replicate. Three replicates were used to represent the result from each cooking treatment. Broccoli was cooked for 2 min and 5 min using each cooking method to evaluate metabolite loss as a function of cooking time. For all three cooking methods, 2-min treatment and 5-min treatment were conducted separately, so no interruption occurred in 5-min treatment.

#### 2.2.2. Boiling Treatment

For boiling treatment, the broccoli florets were immersed in 700 mL boiling water for 2 or 5 min, following which they were immediately removed from the boiling water and transferred to an ice-water bath.

#### 2.2.3. Steaming Treatment

For steaming treatment, the broccoli florets were evenly spread out without any overlap on a stainless-steel steamer basket, which was placed over 700 mL boiling water for 2 and 5 mins. Steamed broccoli was immediately transferred to an ice-water bath for rapid cooling.

#### 2.2.4. Microwaving Treatment

For microwaving treatment, a microwave oven (Whirlpool Corporation, Benton Harbor, MI, USA, 49022) was used at 50% power (maximum power: 1000 W). The broccoli florets were placed on a plate and covered with a paper towel soaked in 50 mL water. The broccoli samples were microwaved for 2 or 5 min and were then transferred to an ice water bath for cooling. All cooked broccoli samples were packed in a zipper bag, which were then placed in a 68-L container (Sterilite, Townsend, MA, USA, 01469) and moved to a −20 °C walk-in cooler prior to lyophilization. 

#### 2.2.5. Cooking Water Volume Adjustjment for Quantification

The volume of the cooking water of the boiling and steaming treatments was adjusted to 700 mL before collecting in 50 mL twist tubes. For the microwaved sample, the damp paper towel was first rinsed in 200 mL water, after which the water was manually squeezed out of the paper towel and the volume was adjusted to 700 mL. All cooking water samples were stored in a 4 °C walk-in cooler for further analysis.

### 2.3. Quantification of Glucosinolates (GSLs) via UHPLC-DAD

GSLs in samples were analyzed using a previously published method [28] with slight modifications. A quantity of 75 mg of freeze-dried powdered material was weighted in a 2-mL screw top minicentrifuge tube. The samples were then extracted with 0.75 mL 70% aqueous methanol and placed onto a 95°C shaking heating block for 10 min. After 10 min incubation, samples were cooled down on the ice for 5 min and 187.5 µL internal standard (0.907 mM glucosinalbin, isolated from *Sinapis alba*) was added to each sample. The samples were vortexed for 10 s and then centrifuged at 12,000× *g* for 10 min. The supernatant was collected in a new 2-mL microtube and stored on ice. The pellet was then re-extracted using the same condition described above, and the supernatant was collected into the corresponding 2-mL microtube from the last extraction. The mixed supernatant was then vortexed for 10 s. The proteins in the supernatant were precipitated with 0.15 mL of a 1:1 mixture of 1 M lead acetate and 1 M barium acetate and centrifuged at 12,000× *g* for 3 min. DEAE Sephadex A-25 resin (GE Healthcare, Piscataway, NJ, USA) was prepared in a ratio of 20 g resin to 100 mL deionized water before the extraction, and 1 mL of well-mixed resin was added into a Poly-Prep column before the desulfation of GSL. After the resin was drained, the supernatant was poured into the Poly-Prep column. The resin was then washed with 3 mL 0.02 M pyridine acetate and then 3 mL deionized water. Once the water eluted out of the column, 500 μL 20 U/mL sulfatase solution was prepared from crude *Helix pomatia* type-1 arylsulfatase powder (Sigma-Aldrich, St. Louis, MO, USA) and incubated for desulfation of GSL overnight at room temperature. After overnight incubation, desulfo-GSL was eluted into 2-mL microtubes with 1.5 mL deionized distilled water, and a syringe with a 0.2-μm nylon filter was used to filter the extract and move the extract into chromatography vials.

The separation and detection of GSLs was conducted on a Nexera-i LC 2040C ultra-high-performance liquid chromatography (UHPLC) (Shimadzu, Kyoto, Japan) machine coupled with a photo diode array detector (DAD). The DAD was set to monitor the absorbance at 229 nm. The filtered samples were injected into the Kromasil reverse-phase C18 HPLC column (1.8 μm, 100 mm × 2.1 mm i.d., 100 Å) at 40 °C. The selection of mobile phase was deionized distilled water for solvent A and 100% acetonitrile for solvent B under the flow rate of 1 mL per minute, and gradient elution was conducted: from 0% B at 0 min to 4% B at 7 min, 20% B at 20 min, 25% B at 35 min, 80% B at 36 min, then 80% B at 40 min. The system was set to 0% B at 41 min until 50 min to reset the column condition for the next injection. Relative response factor (RRF) was adapted for quantification and the quantification was based on the internal standard, glucosinalbin, of which the RRF was 0.5 [29]. For other detected GSLs, the determined RRF values of glucoiberin, progoitrin, glucoraphanin, sinigrin, gluconapin 1.11, glucoerucin 1.00, glucobrassicin 0.29, 4-methoxyglucobrassicin 0.25, gluconasturtiin neoglucobrassicin, 1-hydroxyglucobrassicin/4-hydroxyglucobrassicin were 1.07, 1.09, 1.07, 1, 1.11, 1.00, 0.29, 0.25, 0.95, 0.20 and 0.28, respectively [29,30,31]. The presence of 1-hydroxyglucobrassicin in *Brassica* plants was rarely reported [30,32], but we do not want to exclude the possibility of 1-hroxyglucobrassicin in the sample because we detected significantly higher transcript abundance of CYP781F4 under the same treatment condition in ‘Red Russian’ kale using the primers designed with the published sequence information of the *Brassica oleracea* database [33]. 

To tentatively identify the desulfo-GSL present in the sample, a LC-tandem mass spectrometry (MS) system Waters 32 Q-Tof Ultima spectrometer coupled to a Thermo Accela 1200 UHPLC system coupled to a heated ESI source and to a Q Exactive high-resolution (HR) quadrupole and orbitrap LC-MS/MS (Thermo Scientific, Waltham, MA, USA) was employed. Identification information was obtained by comparing the previous publication from our lab [30], the fragmentation diagnostic ions presented in Kusznierewicz’s work [34], and the information from Clarke’s previous work [29]. 

GSLs in cooking water was determined using the same protocol mentioned above with slight modification. Precipitation of protein in GSL cooking water was completed with the addition of 0.37 mL of a 1:1 1 M barium acetate: 1 M acetate acetate mixture and followed the same steps described in the last section.

### 2.4. Quantification of GSL Hydrolysis Products via GC-MS

50 mg freeze-dried broccoli powder was weighed in a 2-mL Teflon tube (Fisher Scientific, Waltham, MA, USA) and 1 mL distilled water was added for GSL hydrolysis. Tubes were stored inside a drawer to prevent light exposure. Under this setting, GSL hydrolysis products were generated by endogenous myrosinase at room temperature for 24 h. To capture GSL hydrolysis products, 1 mL dichloromethane was added after a 24 h reaction, and the samples were vortexed and centrifuged after 2 min at 12,000× *g* to separate the liquid into aqueous and organic layers. After the centrifuge, the lower layer (dichloromethane) was pipetted out into a 1.5-mL vial and analyzed using Trace 1310 gas chromatography (GC) (Thermo Fisher Scientific, Waltham, MA, USA) with a single quadrupole MS detector system (ISQ QD, Thermo Fisher Scientific, Waltham, MA, USA) and an autosampler (Triplus RSH, Thermo Fisher Scientific, Waltham, MA, USA). Samples were injected into a 30-m Rxi-5Sil MS capillary column (0.25 mm, 0.25 m, w/10 m Integra-Guard Column; Restek, Bellefonte, PA, USA) for analyte separation. The temperature of the injector was set at 270 °C and the temperature of the detector was 275 °C. The GC temperature programming was set as below: initial temperature (40 °C) was held for 2 min and the oven temperature was increased to 320 °C at a 15 °C min^−1^ rate. When the oven reached 320 °C, this temperature was held for another 4 min to slightly bake the column to wash out remaining analytes and prepare for the next injection. The flow rate of the carrier gas, helium, was set at 1.2 mL per minute. Identification of detected MS fragments was based on the information the National Institute of Standards and Technology (NIST) library or previous publications [35]. 

### 2.5. Quantification of Myrosinase Activity and Nitrile Formation via GC-MS

Myrosinase activity and nitrile formation were measured to estimate the interaction between epithiospecifier protein (ESP) levels and epithiospecifier modifier 1 (ESM1) based on published methods. Myrosinase activity was calculated as the total hydrolysis product hydrolyzed from GSLs within 60 min of incubation. One unit is defined as 1 µmol total GSL hydrolysis product released per minute. Nitrile formation was conducted by incubating crude protein extracts from broccoli samples with concentrated horseradish root extract. Horseradish extract was utilized in this analysis because it served as a highly saturated exogenous substrate source of sinigrin and gluconasturtiin GSL profile of horseradish, and it can minimize the reaction of endogenous myrosinase in broccoli with endogenous GSL substrates [36,37].

A quantity of 75 mg freeze-dried broccoli powder was weighed in a 2-mL microcentrifuge tube and then added to 1.5 mL concentrated horseradish root extract. The samples were vortexed for 10 s and then centrifuged at 12,000× *g* for 5 min. A quantity of 0.5 mL of the supernatant was transferred to 1.5-mL Teflon centrifuge tubes (Savillex Corporation, Eden Prairie, MN, USA) with the addition of 0.5 mL before the incubation. To prevent the samples from potential light exposure, incubation was conducted in the drawer for 10 min at room temperature. After the incubation, samples were vortexed and centrifuged at 12,000× *g* for 4 min and the organic layer (dichloromethane) was collected into chromatography vials. Samples were then analyzed by the GC-MS system with the same thermal programming (in Section 2.4) for GSL hydrolysis product profiling. The calibration curves of allyl isothiocyanate, 2-phenthyl isothiocyanate, and 3- phenylpropionitrile (Sigma-Aldrich, St Louis, MO, USA) were used for quantification. The calibration curve of allyl isothiocyanate was also applied to quantify 1-cyano-2,3-epithiopropane.

### 2.6. Measurement of Electrical Conductivity in Cooking Water

Electrical conductivity (EC) was measured using TechPro II™ (Myron L^®^, Carlsbad, CA, USA). The probe was calibrated first with deionized water. Deionized water was also used to rinse out the sampling probe thrice between measurements and the sampling probe was dried using Kim wipes before the measurement. EC was measured thrice for each cooking water sample and the averaged value of each sample was used for statistical analysis.

### 2.7. Untargeted Primary Metabolites by GC-MS

GC-MS, as described in Section 2.4, was also used to determine the primary metabolite profile in broccoli samples with derivatization. Primary metabolites were extracted using a previously published study [38] with slight modifications. A quantity of 50 mg of broccoli lyophilized powder was weighed in 2-mL microcentrifuge tubes and 1.4 mL methanol was then added into the samples for extraction. A quantity of 80 μL of internal standard (10 mg/mL ribitol) was also added into the samples. After 10 s of vertexing, the extracts were centrifuged and the supernatants were poured into a new set of 2-mL micro tubes. Quantities of 375 μL of −20 °C chloroform and 700 μL 4 °C water were added into samples for polar compound fractionation. The extracts were then vortexed for 20 s, then centrifuged and transferred to 1.5-mL tubes for drying. The drying procedure was completed in a Vacufuge™ concentrator (Eppendorf, Thermo Fisher Scientific, Waltham, MA, USA). After drying, samples were derivatized by incubating the extract with 50 μL of 40 mg/mL methoxyamine hydrochloride in pyridine for 90 min on a 37 °C heating block. After this incubation, 70 μL *N*-Methyl-*N*-(trimethylsilyl) trifluoroacetamide (MSTFA) with 1% trimethylchlorosilane (TMCS) was added into the sample for another 30-min incubation on a 37 °C heating block. Extracted analytes were then injected into the same GC-MS system mentioned in Section 2.4 for profiling. The MS detection used positive electron impact mode (EI) with a *m*/*z* 40–500 scan range and ionization energy at 70.0 eV.

After the GC-MS analyses were completed, raw files generated from primary metabolite profiling were converted to mzXML file format by RawConverter [39]. The detection and alignment of extracted peaks were completed using the XCMS package in R studio with default settings [40]. In R multivariate or univariate statistics by MetaboAnalyst [40], data underwent Pareto scaling and was normalized to a unique ion peak (319 *m*/*z*) generated from the spiked internal standard (ribitol). Tentative metabolite annotation was done depending on the fragment pattern and retention times generated by authentic standards using the same chromatogram conditions or comparing with the mass spectra present in the NIST library.

### 2.8. Univariate and Multivariate Analyses

JMP 14 (SAS Institute, Cary, NC, USA) was used for the statistical analyses. Two-way univariate analysis of variance (two-way ANOVA) was used to determine the effect of MeJA treatment and cooking method on the metabolites. Data was separated into (A) raw and 2-min cooking and (B) raw and 5-min cooking when two-way ANOVA was used. A slice test was used to determine the effect of MeJA on metabolites when the interaction between two factors (MeJA treatment and cooking method) was significant at *p* ≤ 0.05. If no interaction was detected, a Student’s T-test was used to determine the effect of MeJA on metabolites under the same cooking method, and the significance was *p* ≤ 0.05. 

For primary metabolite analysis, the acquired chromatograms, converted to mzXML, were used to conduct partial least-square discrimination analysis, variable importance in projection value estimation, and ANOVA in MetaboAnalyst [40]. 

## 3. Results and Discussion

### 3.1. Effect of MeJA and Cooking Method on Glucosinolate Profile

A total of 10 GSLs were detected in broccoli samples, and the results of two-way ANOVA (Appendix A) showed a significant interaction (*p* ≤ 0.05) between MeJA treatment and cooking method in total indole GSLs and total GSL, irrespective of the cooking time. The concentration of total GSL in MeJA-treated broccoli was significantly higher (*p* ≤ 0.001) in MeJA-treated broccoli among all cooking methods (Figure 1). The concentration of total GSL in raw, steamed and microwaved MeJA-treated broccoli was the greatest across all samples by post-hoc Tukey’s HSD test at *p* ≤ 0.05, with no significant difference between each other. Boiling led to the maximum loss of GSL among all MeJA-treated broccoli, which corresponded well with previous studies [10,11]. However, the concentration of glucoraphanin was significantly reduced (*p* ≤ 0.05), which was different from a previous study [6] that found that the concentration of glucoraphanin did not differ between control and 500 µM MeJA-treated broccoli [6]. The decrease may occur during postharvest, since glucoraphanin was rapidly reduced in MeJA-treated broccoli 10 days after the harvest when stored at 4 °C [6]. 

Regardless of different cooking methods and durations, the total GSL amount in MeJA-treated broccoli was still higher than in the non-treated broccoli. This suggests that the increased GSL concentration in broccoli samples was solely affected by MeJA treatment, and the effect of MeJA was not affected by cooking methods. Reports show that the change in GSL during the cooking process can depend on GSL structure [41], plant matrix [42], or cellular environment [43]. The use of larger volumes of cooking water (boiling) also leads to a higher loss of GSL [44]. GSL loss during the cooking process may be attributed to cell lysis and thermal degradation [41]. GSL concentration in the cooking water was measured and varied between cooking methods. The greatest concentration of total GSL in cooking water was detected in 2-min boiling water from MeJA-treated broccoli (Appendix A) and no interaction of MeJA treatment and cooking method was detected. Results were consistent with previous literature where boiling was reported to lead to a great loss of GSL [41,45]. Electrical conductivity (EC) of the cooking water was also measured (Appendix A; Figure 2) but interactions from the two factors (MeJA treatment and cooking methods) were not detected (*p* = 0.48 for 2-min cooking of broccoli and *p* = 0.12 for 5-min cooking of broccoli). In 5-min boiling and steaming cooking treatment, the EC value of the cooking water from MeJA-treated broccoli is significantly lower than that from the control broccoli. From the two-way ANOVA analysis for the 5-min cooking experiment, the effect of MeJA treatment on the EC value was significant, while there was no significant effect on EC in cooking water from 2-min cooking (Appendix A). The EC value of the boiling water of MeJA-treated and untreated broccoli was also the highest among all the samples, which suggested a high level of cell lysis [10] and leaching of GSL into the cooking water. 

MeJA treatment might change the cellular environment [46] and alter cell structure by changing the cell wall composition [47,48,49], which affects leaching of GSL from the cell. In this study, total GSL content in MeJA-treated broccoli was 1.6-fold higher than in untreated control broccoli after 5-min boiling, which was considered the most disruptive cooking method in this study (Figure 1). Furthermore, total GSL amount in MeJA-treated broccoli was significantly higher than in all steamed samples and in the 5-min microwaving groups compared to the untreated broccoli. For milder cooking methods (2-min steaming or 2-min microwaving), the concentration of total GSL in MeJA-treated broccoli was 2.5-fold higher than in the untreated control broccoli. Many studies have suggested that intact GSL in food can be hydrolyzed via the myrosinase-like enzyme produced by the human gut microbiota [50], while the efficiency may depend on the individual and the types of microbes. Moreover, a 2020 study from Liou et al. has revealed the bacteria species that convert GSL into isothiocyanates in human gut [51]. 

Although health-promoting effects of indole GSL and its hydrolysis products are reported, it was also suggested that neoglucobrassicin may pose risks to humans. Neoglucobrassicin in pok choi (treated with 2 mM MeJA) sprout juice was reported to have high mutagenic activity using human sulphotransferase (SULT) 1A1 (hSULT1A1) [24]. I3C, a hydrolysis product from glucobrassicin, was also reported to have Janus properties in animal models [25]. Considering that anticarcinogenic and carcinogenic effects of indole GSLs and hydrolysis products have both been reported previously, the GSL profile provided in this study may be important for dieticians and professionals once a better understanding is reached. If a certain concentration of neoglucobrassicin is determined to be health-promoting in the future, then MeJA treatment can be a valuable tool to deliver high GSL into the human digestive system via *Brassica* crops as a vehicle. 

### 3.2. Effect of MeJA Treatment and Cooking Method on Myrosinase Activity and GSL Hydrolysis Products in ‘Green Magic’ Broccoli 

In raw broccoli, 250 µM MeJA treatment significantly increased myrosinase activity by 37% (Figure 3). We also indirectly measured epithiospecifier protein (ESP) activity by incubating broccoli crude protein and horseradish GSL extract together [36]. Nitrile formation (%) of gluconasturiin was reduced, although the result was not statistically significant (*p* = 0.06, not presented). 

Eleven GSL hydrolysis products were detected in raw broccoli, including ITC, nitriles, indoles and oxazolidine-thione (Appendix A). With the substantial increase in neoglucobrassicin and glucobrassicin after MeJA treatment (Appendix A), the levels of hydrolysis products derived from neoglucobrassicin (NMI3C, *N*-methoxyindole-3-carboxyaldehyde (NMI3CA), *N*-methoxyindole-3-acetonitrile (NMI3ACN)), and indole-3-carbinol (I3C) from glucobrassicin were significantly higher (*p* ≤ 0.05) in raw MeJA-treated broccoli than in untreated broccoli. NMI3C and I3C were reported to have anti-inflammatory and chemopreventive effects [23,52]; moreover, NMI3C may be a stronger inhibitor than I3C of tumor development [53]. Therefore, the increase in the levels of these compounds after MeJA treatment might improve the nutritional value of raw broccoli. NMI3C and NMI3ACN levels were significantly higher in MeJA-treated broccoli for most of the cooking methods (Appendix A). However, other GSL hydrolysis products were not affected by the MeJA treatment (Appendix A); therefore, the effect of MeJA on hydrolysis products may vary with precursor GSL. 

In Figure 4, the abundance of GSL hydrolysis products in cooked samples was lower compared to the raw samples. This result corresponds well with the myrosinase activity decrease in the cooked samples (Figure 3). As the abundance of GSL hydrolysis produces contributes significantly to broccoli’s taste and determines consumer acceptance [54], it is plausible that cooking treatment of MeJA-treated broccoli may be more acceptable to average consumers, while these broccolis still contain a high amount of GSL before cooking. 

Myrosinase activity was close to zero in all the cooked samples (Figure 3). Myrosinase inactivation during the cooking process may reduce the content of all hydrolysis products, as no significant differences in hydrolysis product levels were observed between the majority of cooked MeJA-treated and untreated broccolis. Therefore, although MeJA treatment might increase myrosinase activity in the raw material, the effect was not observed after 2-min boiling, steaming, or microwaving [53]. We recently found that MeJA-treated raw broccoli significantly reduced consumer preference, but treatment did not significantly change consumer preference after cooking because of the low concentration of hydrolysis products from neoglucobrassicin by myrosinase inactivation [54]. Thus, MeJA-treated broccoli can be favorable delivered as frozen broccoli or cooked broccoli rather than raw. Previous results reported that there is very low or no myrosinase activity in frozen broccoli, which does not form sulforaphane [55], but it does not produce an off-flavor from neoglucobrassicin in this case.

### 3.3. Effect of MeJA and Cooking Method on Primary Metabolites in ‘Green Magic’ Broccoli with or without MeJA Treatment 

Studies have shown that MeJA may change primary metabolites, including sugar, amino acids, and organic acids, by regulating the balance between growth and defense activities [30,56]. These primary metabolites can be valuable nutrition components; however, the effect of MeJA treatment on primary metabolites in cooked broccoli has not been studied. Partial least-square discrimination analysis (PLS-DA) (Figure 5) was used to identify potential biomarkers with variable importance in projection (VIP) values over 1.5, which indicates that this metabolite contributed greatly to the differences between groups (Appendix A). 

In the raw broccoli, MeJA treatment significantly reduced (*p* ≤ 0.05) oxoproline, glutamic acid, *myo*-inositol, and sucrose content (Table 1). This was consistent with published studies where MeJA treatment reduced sugars and amino acids in other *Brassica* vegetables [35,57]. Therefore, MeJA-mediated reduction in sugar and amino acid content was common for many *Brassica* species. The VIP values of primary metabolites, including sucrose, glucose, fructose, *myo*-inositol, oxoproline, and quininic acid, were consistently high (>1.5) with all cooking methods; hence, changes in the levels of these compounds were due to the effect of MeJA and not because of cooking methods. In other words, 250 µM MeJA treatment changed the amount of individual primary metabolites in raw broccoli, and this effect was still observed after all cooking methods. 

MeJA treatment significantly increased the amount of quinic acid in the 2-min boiling, 2- and 5-min steaming, and 2- and 5-min microwaving samples. The levels of amino acids (oxoproline, glutamic acid, valine, isoleucine, proline, serine, and alanine) were significantly reduced by MeJA treatment, and those of glutamic acid, proline, and serine were reduced by >50% (Table 1). The amounts of sucrose and *myo*-inositol were significantly lower in all MeJA-treated cooked broccoli; however, the amounts of glucose and fructose showed opposite trends. MeJA-treated broccoli contained significantly higher amounts of glucose after 2 min boiling and microwaving. In addition, the amount of fructose was significantly higher after 5 min microwaving of MeJA-treated broccoli. 

It has been reported that amino acids can taste sweet, sour, bitter, or umami [58]. For example, glutamate is sometimes used as an umami ingredient in culinary practices. In this study, amino acids associated with bitterness (valine and isoleucine), sweetness (alanine, proline, and serine), and umami (glutamic acid) were all significantly changed by 250 µM MeJA treatment. Sugars, such as sucrose, are associated with masking the bitterness of *Brassica* vegetables [59], and glucose correlates highly with the perception of sweetness [59]. In fact, 250 µM MeJA treatment on ‘Green Magic’ broccoli was found to change the sensorial perception of raw broccoli but not of cooked broccoli, as determined by a consumer panel. The perceptible changes were mainly attributed to neoglucobrassicin and its hydrolysis products, not to amino acids or sugars [54]. Understanding the changes of primary metabolites in response to MeJA treatment and to various cooking methods may provide valuable insights regarding the nutritional and sensorial values of broccoli. 

## 4. Conclusions

In this study, broccoli treated with 250 µM MeJA contained significantly greater concentrations of GSL than untreated broccoli even after cooking by boiling, steaming or microwaving. Exogenous 250 µM MeJA application will increase the nutritional value of cooked broccoli because of the increased concentration of GSL, without producing bad flavors from GSL hydrolysis products even after cooking. MeJA-treated broccoli has potential for use as a value-added ingredient and, since MeJA can be applied to other *Brassica* vegetables, this treatment may be used to enhance the nutritional value of other commodities once the health-promoting effects of indole GSL and the hydrolysis products are better understood. 

## Figures and Tables

**Figure 1 foods-09-00758-f001:**
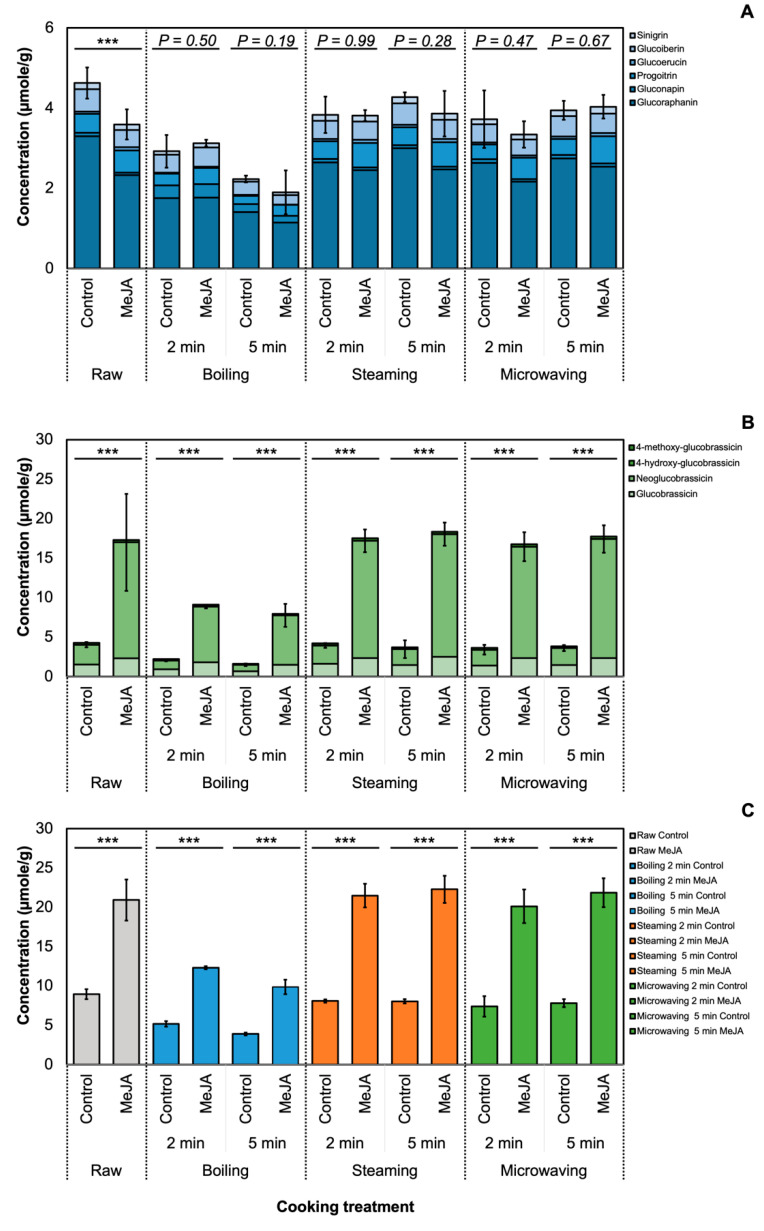
Effect of cooking and 250 µM MeJA treatment on total aliphatic glucosinolates (**A**), total indole glucosinolates (**B**), and total glucosinolates (**C**) in ‘Green Magic’ broccoli. C: control broccoli; M, MeJA-treated broccoli. Asterisk (*) = detected significant different by Student’s T-test (*p* ≤ 0.05, *n* = 3) with a significant interaction between MeJA treatment and cooking treatment detected (Appendix A). ***, *p* ≤ 0.001.

**Figure 2 foods-09-00758-f002:**
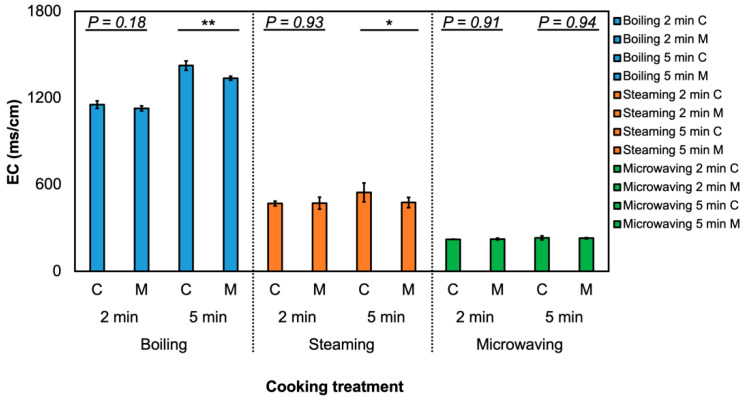
Electrical conductivity in cooking water. Asterisk (*) indicates significant difference with or without MeJA within the same cooking treatment by Student’s T-test (*p* ≤ 0.05, *n* = 3) ^ns^, *, *p* ≤ 0.05; **, *p* ≤ 0.01. The first letter, C or M, indicates the control or methyl jasmonate-treated group. The second letter, B, M, or S, indicates different cooking treatments including boiling, microwaving, and steaming. The last number after the B, M, or S indicates the duration of the cooking method (mins).

**Figure 3 foods-09-00758-f003:**
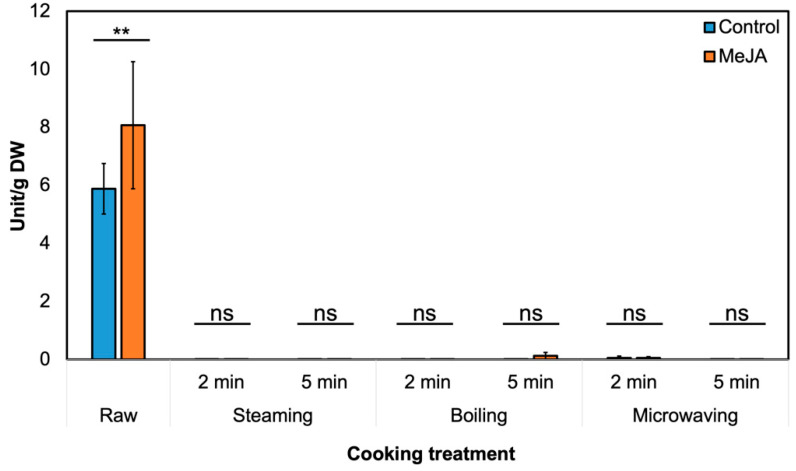
Effect of cooking and 250 µM MeJA treatment on myrosinase activity in broccoli. Asterisk (*) = detected significant difference with MeJA treatment by Student’s T-test at *p* ≤ 0.05 (*n* = 3). One unit = 1 µmole of total released GSL hydrolysis products in 1 minute. ^ns^, not significant; **, *p* ≤ 0.01.

**Figure 4 foods-09-00758-f004:**
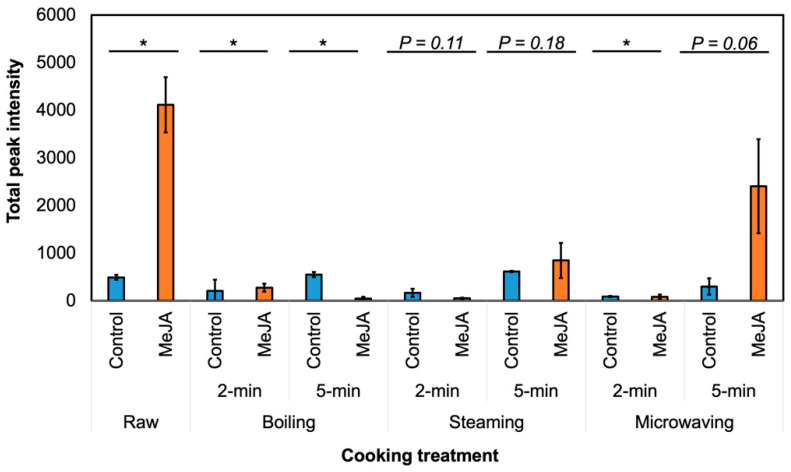
Total peak intensity of glucosinolate hydrolysis products in raw and cooked broccoli. MeJA, broccoli with 250 μM methyl jasmonate treatment. Asterisk (*) = significant difference with or without MeJA by Student’s T-test (*p* ≤ 0.05, *n* = 3) under the same cooking conditions. ^ns^, *, *p* ≤ 0.05.

**Figure 5 foods-09-00758-f005:**
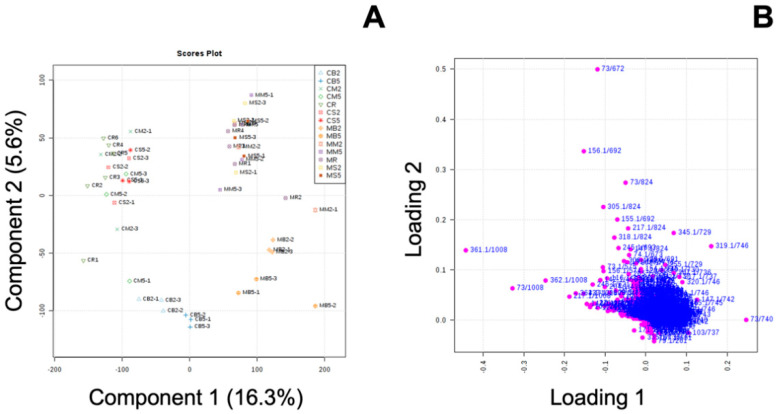
Partial least-square discrimination analysis score plot (**A**) and loading plot (**B**) derived from raw (control raw: CR; methyl jasmonate-treated raw: MR) and cooked control and methyl jasmonate-treated broccoli samples. The first letter, C or M, indicates the control or methyl jasmonate-treated group. The second letter, B, M, or S, indicates different cooking treatments including boiling, microwaving, and steaming. The last number after B, M, or S indicates the duration of the cooking method (mins).

**Table 1 foods-09-00758-t001:** Fold changes of primary metabolites in MeJA-treated ‘Green Magic’ compared to untreated broccoli under the same cooking treatment.

Cooking Method		Amino Acids ^†^							Sugar and Sugar Derivatives				Organic Acids
		Oxo- Proline	Glu	Val	Pro	Ser	Ile	Ala	Fructose	Glucose	Sucrose	*myo*-Inositol	Quinic Acid
Raw		0.58 *	0.42 *						0.94	1.06	0.28 *	0.91 *	1.35
Boiling	2 min	0.77 *	0.35 *						0.96 *	1.09 *	0.26 *	0.90 *	1.35 *
	5 min	0.95	0.43 *						0.80	1.09	0.37 *	0.84	1.73
Steaming	2 min	0.71 *	0.40 *	0.49 *	0.40 *	0.39 *	0.38 *			1.05	0.32 *	0.74 *	1.72 *
	5 min	0.70 *	0.35 *	0.43 *	0.35 *	0.40 *	0.26 *			0.90 *	0.26 *	0.59 *	1.54 *
Micro- waving	2 min	0.46 *	0.43 *	0.48	0.37 *	0.36 *		0.69 *	0.98	1.05 *	0.26 *	0.76 *	2.04 *
	5 min	0.62 *	0.57 *	0.57 *	0.57 *	0.56 *		0.86 *	1.31 *	1.38*	0.49 *	1.02	2.33 *

† Abbreviations of amino acids: (1) Glu = glutamic acid, (2) Val = valine, (3) Pro = proline, (4) Ser = serine, (5) Ile = isoleucine, and (6) Ala = alanine. Asterisk (*) = detected significant difference with or without MeJA treatment by Student’s T-test at *p* ≤ 0.05 based on the peak intensity (*n* = 3).

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
