# Peer review of "Methyl Jasmonate Treatment of Broccoli Enhanced Glucosinolate Concentration, Which Was Retained after Boiling, Steaming, or Microwaving"

_foods, 2020, doi:10.3390/foods9060758_

Round 1

Reviewer 1 Report

The reviewed manuscript provides updated information on the change in GLS content and their related hydrolysable compounds in Brassica oleracea (var. italic) plants, after the treatment with MeJA. The study also include a comparative evaluation between MeJA and cooking treatments.

The strong points of the revised manuscript are linked to the chemical analysis performed via UHPC-DAD and via GC-MS. Furthermore, the correct use of statistical analyses provides an additional value to the work. The manuscript is very well written, and its structure is clear, intuitive and easy to understand. The introduction includes all the topics related to state of the arts. Moreover, the subject matter is original, and the discussion completely supports the obtained data.

However, a number of small changes should be listed:

In the abstract are reported GLS and Meja, but is not before explained their meaning. Please, provide their full name in extension not only in the introduction but also in the abstract.

I suggest the authors to divide the section 2.2. in 4 small subsection (ex. 2.2.1. preparation; 2.2.2. boiling treatments; 2.2.3. etc….). Moreover, what is reported at the line 105-109 should be moved to correspective subsections.

I suggest to modify the title of the section 2.3 and 2.4 including the method used for the quantification (i.e. quantification of glucosinolates (GLS) via UHPL-DAD).

Figures: The figures are very clear, however Foods does not require additional costs for the coloured figures. For this reason I suggest the authors to re-prepare the figures using different colours for the different treatments. The letters of the panel, when present, should be positioned at the top right of the graph, and outside of them. The brackets are not necessary. Neither reporting the title of the panels is necessary, it is already sufficiently explained in the caption. Finally, the name of the panel in the caption should be reported after the explanation (i.e. Partial least square-discrimination analysis score plot (A) and loading plot (B) derived from……). This problem is also present in all the captions.

Author Response

Thank you very much for your insightful opinions.

Reviewer 2 Report

Review of Chiu et al manuscript 2020 FOOD on neoglucobrassicin levels, MeJa and cooking

The paper is relevant and generally well written. I have numerous technical comments, of which a few are of nutritional importance (notably taking into account known negative effects of indole GSLs in text and last sentence of abstract) and many are of relevance for the data quality.

Title

Methyl Jasmonate Treatment of Broccoli Enhanced Glucosinolate Concentration and they were retained even after Various Cooking Treatments

The scientific content this title is okay, but language-wise I have problems. What about one of these:

Methyl Jasmonate Treatment of Broccoli Enhanced Glucosinolate Concentration also after Various Cooking Treatments

Methyl Jasmonate Treatment of Broccoli Enhanced Glucosinolate Concentration, which was retained even after Various Cooking Treatments

Methyl Jasmonate Treatment of Broccoli Enhanced Glucosinolate Concentration also after Steaming, Boiling or Microwaving

Methyl Jasmonate Treatment of Broccoli Enhanced Glucosinolate Concentration also after Cooking

Personally, I prefer one of the last two for maximum precision or brevity.

Abstract:

Exogenous methyl jasmonate treatment will increase the levels of neoglucobrassicin and their bioactive hydrolysis products in broccoli (Brassica oleracea var. italica), although the fate of MeJA-induced glucosinolates after various cooking methods is unknown.

please change to

Exogenous methyl jasmonate treatment was known to increase the levels of neoglucobrassicin and its bioactive hydrolysis products in broccoli (Brassica oleracea var. italica), but the fate of MeJA-induced glucosinolates after various cooking methods was unknown.

Abbreviation:

The authors abbreviate glucosinolates (in plural) GLS, but later use the same abbreviation for glucosinolate (in singularis). This is disturbing. The problem can easily be solved by changing to the more accepted abbreviation GSL (sing.) and GSLs (plural). Please do.

GSL-classification (line 33-51).

Aliphatic GSLs are not necessarily derived from methionine, not even in Brassica oleracea, since 1-methylpropylGSL is known from the species. Please avoid claiming this. You can change to: In Brassica oleracea, aliphatic GSLs derived from methionine and isoleucine are known (Baek et al., 2016, Klopsch et al., 2018). Aromatic GSLs (line 51) in a chemical nomenclature sense is a very diverse group that by chemical definition includes the indole GSLs (Blazevics et al., 2020). Hence, please rename the group containing gluconasturtiin to benzenic glucosinolates or phenylalanine-derived glucosinolates.

The term “Indolyl GSLs” is often seen but chemically wrong, as the glucobrassicins are indolylmethyl GSLs. This distinction is of practical value after the discovery of true indolyl GSLs in Brassica species after phytoalexin-induction by Prof. Pedras and her group. Hence, please use the generally accepted term indole GSLs when referring to glucobrassicins.

Overall, you must either classify glucosinolates from amino acid precursor only (Met, Ile, Trp, Phe) or change the three categories to aliphatic GSLs, indole-GSLs and benzenic GSLs.

Background on health aspects (line 58-63)

This background only considers positive effects of neoglucobrassicin. However, it is well-established that indole GSLs have “Janus properties” concerning their health effects, with both positive and negative effects depending on the exact setting. Hence, a more balanced background should be given. You can refer to general reviews by e.g. Holst and Williamson and by the George Jander group, as well as newer papers especially from the group around Prof. Monica Schreiner.

The well-established Janus properties should be mentioned in the discussion, and the final sentence of the abstract should be modified accordingly, to take into consideration also possible negative effects of MeJa-induced GSLs.

Data reporting

GSL analytical data must be presented transparently regarding the basis of identification (Blazevics et al., 2020), just like identification of the hydrolysis products was specified in this manuscript, by referring to the libraries used. Since there are no libraries for GSLs, please consider the below.

Please provide a table in the actual article (not supplementary) specifying the glucosinolates identified, their retention time, their mean level in Meja-induced broccoli samples and the basis of their identification. Either conclusive identification based on comparison with a pure authentic standard or a generally accepted reference material such as publicly available rapeseed standards. Or tentative identification based on UV spectrum and comparison to chromatograms in the literature etc. Both kinds of identification are acceptable but must be specified. Please include in the table the sum of any non-identified GSLs or possible GSLs (assuming their RRF to be 1.00), and mention whether such peaks were included or excluded in reported totals. If you decide to disregard non-identified peaks, it is okay, but it must be reported transparently what was done.

Please also illustrate representative chromatograms of both contro; and induced broccoli in the supplementary files, this is much appreciated by readers wishing to understand data-quality.

The method text specifies that 1-hydroxyglucobrassicin was detected (line 130). This GSL has only recently been reported (by Pfalz et al., 2016), and was reported to be an intermediate in the biosynthesis of neoglucobrassicin. Please specify your basis for suggesting this GSLs. This reviewer agrees that it is very likely that 1-hydroxyglucobrassicin should be detectable at trace levels during induction of neoglucobrassicin. Even if your identification of this poorly known GSL was uncertain, please do not hide observations but present any evidence of this biosynthetic intermediate, for the benefit of future investigators. Please include this intermediate even if the level is not distinguishable in the graphs. Please mention briefly in discussion.

In B. oleracea, the aliphatic, isoleucine-derived GSL 1-methylpropylGSL (glucocochlearin) is known in some varieties (Baek et al., 2016, Klopsch et al., 2018). Please specify whether this GSL was detected or possibly detected.

Comments on GSL-analysis

The indicated reference does not contain details of the method, please check the indicated ref 25. Do you mean ref 24?Please double check all reference-numbering. Even if you report another previous reference, please provide all details required in this peer review report. In general, the numerous local conditions used in the GSL analytical literature are of too low transparency.

The authors have used the RRF 1.00 for the internal standard glucosinalbin, and refer to Clarke (2010) for this response factor. However, Clarke cites this RRF to be either 0.4 or 0.50, depending on the source. Please re-calculate all data using the RRF 0.50 for glucosinalbin, as this is the internationally accepted value. The reason for the low RRF is the additional UV chromophore in this GSL.

It is well-established that measured GSL level depend on the purity of the sulfatase used, and most laboratories prefer a purified sulfatase. However, determination is linear both with and without purification. Please specify whether you used the crude enzyme powder from Sigma or a purified fraction. If purification was used (e.g. like Graser et al., 2001), please specify.

The extraction solvent is indicated as 70% methanol. Do you mean 70% aqueous methanol or what were the remaining 30%? The boiling point of 70% aq. methanol is around 70 centigrades. Please reconsider whether your extraction was at 95 centigrades (line 112), if so obtained by closed containers immersed at this temperature, or just “boiling” in an opne container or vial. As GSLs are to some extent thermoinstable, using 95 centigrades for 10 minutes might lower yields significantly compared to 70 centigrades, so this detail is relevant.

It is well-established that correct determination of indole GSLs critically depends on the mass of DEAE-sephadex material used and the elution volume after desulfation (Agerbirk et al., 2001 JAFC). This is because the indole GSLs adher stronger to the DEAE material than e.g. the used internal std. desulfoglucosinalbin, resulting in under-estimation unless a sufficient lution volume is used. Please specify the dry mass of DEAE sephadex material per ion exchange column and the volume of water used for elution.

Supplementary

files are in .doc format. Please convert to pdf for future data integrity.

Author Response

Thank you very much. Please see the attached file for response.

Round 2

Reviewer 2 Report

The revision is mainly satisfactory. I was able to detect a few tiny problems, specified below. The authors should be allowed to correct these as a minor revision, then it is acceptable in my view.

Re-review of cooked MeJa broccoli manuscript:

I still would prefer a representative chromatogram of control broccoli and induced broccoli (or cooked broccoli, to make it even more relevant to this study) in a supplementary figure. But I will not insist. The chromatogram illustrated in the answer to reviewers is satisfactory, so there is nothing to be nervous about in showing it in a supplementary file. Please consider Again whether you would do readers this favour.

Title. I think the revised title is still not grammatically correct. could you change to one of these two:

 Methyl Jasmonate Treatment of Broccoli Enhanced Glucosinolate Concentration also after Boiling, Steaming, or Microwaving

Methyl Jasmonate Treatment of Broccoli Enhanced Glucosinolate Concentration, which was retained after Boiling, Steaming, or Microwaving

or another of the previously suggested options.

abstract

line 21 and its hydrolysis products…

line 24: please delete “patient’s”, since GSLs are mainly regarded as preventive ingredients, i.e. acting to prevent the development of cancer. That is, you should eat broccoli to avoid becoming a cancer patient…

line 912 and elsewhere including supplementary tables, Student’s t-test is spelled with a non-capitalized t, as far as I am aware.

line 1124-1125: a “)” is missing in the following sentence, where a section with neoglucobrassicin products is started with a “(“, but never ended:

neoglucobrassicin (NMI3C, N-methoxyindole-3-carboxyaldehyde (NMI3CA), N-methoxyindole-3-1124 acetonitrile (NMI3ACN), and indole-3-carbinol (I3C) from glucobrassicin were significantly higher

Since the list of detected products of neoglucobrassicin ends with …(NMI3CACN), a second “)” should be added after this. So the lines should be changed to

neoglucobrassicin (NMI3C, N-methoxyindole-3-carboxyaldehyde (NMI3CA), N-methoxyindole-3-1124 acetonitrile (NMI3ACN)), and indole-3-carbinol (I3C) from glucobrassicin were significantly higher

including a double “))”

section 2.3 headline missing C in HPLC

Author Response

Thank you for your suggestion. 
